# Incidence and Characteristics of Perianal Infections in CPX-351-Treated AML Patients

**DOI:** 10.3390/cancers18020208

**Published:** 2026-01-09

**Authors:** Elisa Buzzatti, Cristina Mauro, Cristiano Tesei, Giovangiacinto Paterno, Raffaele Palmieri, Fabiana Esposito, Elisa Meddi, Federico Moretti, Marco Zomparelli, Lucia Cardillo, Carmelo Gurnari, Luca Maurillo, Francesco Buccisano, Adriano Venditti, Maria Ilaria Del Principe

**Affiliations:** 1Hematology, Department of Biomedicine and Prevention, University Tor Vergata of Rome, 00133 Rome, Italy; 2Hematology and Stem Cell Transplant Unit, IRCCS Regina Elena National Cancer Institute, 00144 Rome, Italy; 3UOSD Mieloproliferative, Policlinico Tor Vergata, 00133 Rome, Italy; 4Department of Translational Hematology and Oncology Research, Taussig Cancer Institute, Cleveland Clinic, Cleveland, OH 44195, USA

**Keywords:** acute myeloid leukemia, CPX-351, perianal infections

## Abstract

Despite CPX-351’s design to minimize gastrointestinal issues, its effect on serious infectious complications, such as perianal infections (PIs), remains uncertain in patients affected by acute myeloid leukemia (AML). Our study revealed a notably high incidence of PIs in our secondary AML cohort that underwent CPX-351. The development of a PI was strongly linked to a significantly longer hospital stay and correlated with signs of mucosal damage and the presence of multidrug-resistant organisms, particularly Klebsiella pneumoniae. While patient outcomes concerning early mortality were favorable, the substantial morbidity and prolonged hospitalizations underscore the clinical burden of PIs. This suggests that implementing routine rectal swab surveillance may be a valuable approach for identifying high-risk patients and guiding preemptive interventions.

## 1. Introduction

Perianal infections (PIs) are a complication affecting around 7% of acute leukemia patients [1,2]. These infections encompass perianal abscesses and anal fistulas. A perianal abscess presents as a tender, erythematous, firm, or fluctuant mass near the anus. Conversely, an anal fistula is a tunnel connecting an opening in the skin near the anus to the rectal canal. Several factors contribute to the heightened risk of developing PIs in hematologic patients: neutropenia, mucositis causing damage to mechanical barriers, the inherent compromised immunity, and the necessity for repeated, long-term chemotherapies [3,4]. In the literature, the course of the neutrophil count during the post-chemotherapy phase has consistently been considered a critical factor, influencing both the severity of PIs and, indirectly, the choice of antibiotic treatment and overall prognosis [5,6]. PIs in this patient population are a serious concern, as they can progress from localized issues to life-threatening systemic complications, including septic shock and mortality [7].

Patients with acute myeloid leukemia (AML) face an increased risk of recurrent PIs compared to those with acute lymphoblastic leukemia [7]. This susceptibility is largely attributed to the high-dose cytarabine-based chemotherapies typically administered for AML, which induce more profound neutropenia and mucositis, along with a historical diminished reliance on growth factors.

CPX-351 is a new intensive chemotherapy regimen specifically tailored for patients with secondary AML. This innovative treatment is a dual-drug liposomal encapsulation of cytarabine and daunorubicin, which delivers a synergistic 5:1 drug ratio directly to leukemia cells, significantly sparing normal bone marrow in the process [8].

Due to its unique mechanism, CPX-351 promises gastrointestinal benefit over the classic “7 + 3” regimen, specifically by helping to maintain a functional mucosal barrier and microbiota, reducing intestinal inflammation [9].

Despite these encouraging effects on the gut, it remains an unanswered question whether this translates to a reduced occurrence or altered presentation of PIs. Therefore, specific data on the incidence and characteristics of PIs in patients undergoing CPX-351 therapy is crucial, as current information remains limited.

## 2. Materials and Methods

### 2.1. Data Collection

We retrospectively collected data about the first documented occurrence of PIs in all adult patients diagnosed with secondary AML (sAML), according to WHO 2016 [10], and receiving intensive induction chemotherapy with CPX-351 at the Hematology Unit of Policlinico Tor Vergata Hospital between May 2020 and July 2025. Cumulative incidence was calculated as the number of patients who developed a first PI during CPX-351 treatment divided by the number of CPX-351–treated secondary AML patients at risk at the start of the observation period. Clinical and laboratory data were extracted from an anonymized database. The study protocol was approved by the local Ethics Committee and conducted in accordance with the Declaration of Helsinki.

Analyzed variables included the following baseline parameters: sex, age, number and type of comorbidities, duration of neutropenia and length of hospital stay, time to next chemotherapy cycle, use of parenteral nutrition, and granulocyte colony-stimulating factor (G-CSF) administration, whether the patient was receiving antibiotic or antifungal prophylaxis, and the results of routine rectal swabs. For patients who developed PIs, we collected detailed clinical information, including infection characteristics as the type of infection, grading of severity according to Common Terminology Criteria of Adverse Events (CTCAE) version 5.0., the microbiological results of the lesion swab, hematological parameters, management strategies (details on empirical or targeted antibiotic therapy, whether a surgical approach was undertaken) and infection outcomes (infection-related mortality or shock, the status of the infection at 30 and 60 days post-diagnosis, occurrence of PIs recurrence, and whether the infection led to delays in subsequent chemotherapy cycles). For pain assessment, we used the numeric rating scale (NRS) [11], a numerical scale ranging from 0 (no pain) to 10 (worst imaginable pain). Pain intensity was recorded daily from the onset of the PI until its resolution, and for each patient, the mean NRS value over this period was calculated and used for analysis.

The infection status was defined as resolved upon the complete disappearance of the lesion and associated pain; improved with a reduction in both lesion size and patient pain; stable if the lesion size and pain remained unchanged; and worsened when an increase in both was observed. The study was approved by the local institutional review board (Comitato Etico Territoriale Lazio Area 2, RS 25.25) and conducted in accordance with the ethical principles outlined in the Declaration of Helsinki.

### 2.2. Statistics

Descriptive analyses were performed on the entire study cohort and stratified by study group. The Shapiro–Wilk test was used to assess the normality of each continuous variable. Continuous variables are expressed as mean ± standard deviation (SD), and categorical variables are expressed as numbers and percentages. Differences between groups were evaluated using parametric and non-parametric tests (Pearson’s chi-square test for categorical variables and the Wilcoxon test and linear model ANOVA for continuous variables). When applying logistic regression models to binary data in small populations, conventional logistic regression based on standardized maximum likelihood estimation can encounter several problems. One of the suggested methods to overcome these problems is to use Firth’s logistic regression or penalized logistic regression. Penalized logistic regression enables convergence to finite estimates under separation conditions [12,13]. In our study, Firth’s logistic regression was used to evaluate the relationships of factors predicting perianal infection, given the small population size. To report these relationships, we used a Firth’s multivariate logistic regression odds ratio (OR) and a 95% confidence interval (CI). A significance level of *p* < 0.05 was established. All analyses were performed using RStudio 2025.05.1 + 513 Posit Team (2025). RStudio: Integrated Development Environment for R. Posit Software, PBC, Boston, MA, USA, URL).

## 3. Results

During the study period, 92 AML patients were diagnosed and deemed suitable for intensive chemotherapy. Of these, 22 patients (24%) had sAML and were treated with CPX-351. The cohort exhibited a male predominance (64%) and had a mean age of 58.8 ± 8.6 years (range 43–72 years). The majority of the patients (81.8%) presented with at least two comorbidities, most frequently cardiac (21%) and oncologic (19.5%). A single patient had concurrent inflammatory bowel disease (Crohn’s disease/rectocolitis). Antifungal prophylaxis was universally administered with posaconazole; however, no patients received antibacterial prophylaxis, consistent with institutional guidelines. G-CSFs were administered to 18 patients (81.8%). Parenteral nutrition was required in half of the cohort (50%), primarily indicated for mucositis (6 cases, 27.3%), while the remaining recipients received it as supportive care for weight loss (Table 1). Mucositis was intended as chemotherapy-related inflammation and/or lesions of the oral and/or gastrointestinal mucosa.

PIs occurred in 7 out of the 22 patients (31.8%) treated with CPX-351. All infections were uniformly graded 3 according to CTCAE version 5.0. The PIs presented clinically as perianal abscesses in 3 cases (42.9%), fistulas in 1 case (14.2%), and a combination of both in 3 cases (42.9%). A total of nine microorganisms were isolated from wound swabs in 6 patients (85.7% of the PI group). Polymicrobial infections were detected in two patients (28.5%), involving 2 to 3 pathogens each. The isolates were predominantly Gram-negative bacilli (55.5%) and Gram-positive cocci (44.5%). The most common pathogens isolated were *Klebsiella pneumoniae* and *Enterococcus species* (44.5% each). Of significant concern, one *Enterococcus species* isolate was vancomycin-resistant (VRE), and one *K. pneumoniae* isolate was a carbapenemase producer (KPC). Routine weekly rectal swab monitoring revealed that two patients were colonized with *K. pneumoniae* prior to the onset of their infection; this same organism was subsequently confirmed as the causative agent of their PI. Conversely, no patient in the non-PI group had a positive rectal swab during induction therapy hospitalization. Concomitant bacteremia was observed in 2 patients (28.6%), with one developing septic shock (14.2%). At the onset of infection, 71.4% of PI patients were experiencing severe neutropenia (<0.5 × 10^9^/L), with 3 patients having <0.1 × 10^9^/L (42.9%). The mean neutrophil count at infection onset was 1.695 ± 3.814 × 10^9^/L (range 0–10.290). While both groups experienced severe chemotherapy-induced neutropenia, the mean duration of neutropenia did not significantly differ between the PI and non-PI groups (30.3 vs. 37.9 days, *p* = 0.54). Hospital stay was significantly longer in patients who developed PIs compared to those who did not (mean 49.6 vs. 37.7 days, *p* = 0.034). Targeted antibiotic therapy was administered to 6 patients (85.7%), most often using carbapenems or combination regimens (e.g., meropenem–vaborbactam, ceftazidime–avibactam, tigecycline, or metronidazole-based regimens). The mean duration of targeted therapy was 22 ± 10 days (range 13–38 days).

Following surgical consultation, three patients (42.9%) required surgical drainage and seton placement. The first patient was a 54-year-old male who presented with a large 40 × 30 mm perianal fistula and abscess. The infection was caused by the highly resistant *K. pneumoniae* KPC. The condition arose during the recovery phase from chemotherapy-induced neutropenia. Surgical intervention, consisting of seton placement and vacuum-assisted closure, was performed. Crucially, the procedure commenced once his neutrophil count had recovered to >0.5 × 10^9^/L and his platelet count was >20 × 10^9^/L. The lesion showed clinical improvement after 30 days and was completely resolved within 60 days. The second case involved a 66-year-old man who developed a 30 × 20 mm abscess caused by *Enterococcus faecium* VRE. This infection occurred during the neutropenic phase of chemotherapy. Surgical procedure was performed as soon as his neutrophil and platelet counts reached predefined safe levels, leading to a complete recovery within 60 days. The third patient was a 61-year-old man who presented with the largest lesion, a 45 × 30 mm abscess, coupled with concurrent septic shock. The causative agent was *K. pneumoniae*. The infection arose during a prolonged 50-day period of persistent neutropenia. The presence of a platelet count exceeding 20 × 10^9^/L was deemed sufficient for proceeding. The infection was ultimately resolved without sequelae within 60 days.

No significant differences were observed between the surgical and non-surgical groups regarding age, neutrophil count at the onset of infection, or duration of neutropenia. Although the average hospital stay was longer in the surgical group, this difference was not statistically significant (*p* = ns). The following differences were statistically significant (Table 2): patients who underwent surgery reported a significantly higher mean pain score compared to those who did not (*p* = 0.001). A significantly higher proportion of patients in the surgical group achieved a platelet count exceeding 20 × 10^9^/L during hospitalization (*p* = 0.028).

All patients were alive 30 days post-treatment. At 30 days, the infection was reported as resolved in one patient (who remained neutropenic), improved in three patients (two of whom remained neutropenic), and stable in three cases (one of whom remained neutropenic). By 60 days, all PIs had successfully resolved, except for one patient who subsequently died due to respiratory failure; critically, no patient experienced a recurrence of the infection during the entire follow-up period. The time required until the start of the subsequent chemotherapy cycle did not differ significantly between the PI group (mean 83.5 days) and the non-PI group (mean 80.7 days, *p* = 0.8). The univariate analysis revealed that the development of a PI was statistically associated only with a significantly longer hospital stay (*p* = 0.034). Conversely, no significant associations were observed for demographic factors, comorbidities, use of parenteral nutrition, duration of neutropenia, or use of G-CSFs, though two non-significant trends suggested a potential association between PI development and a positive pre-infection rectal swab (*p* = 0.090), and the presence of mucositis (*p* = 0.053) (Table 3).

The adjusted Firth logistic regression model for the association between the prognostic factors of PIs is presented in Table 4. The multiple Firth logistic regression results showed that male patients (OR_male/female_ 6.339; *p* = 0.214) were at a higher risk of PIs than females, previous perianal swab positivity (OR_negative/positive_ 5.554; *p* = 0.391) and mucositis (OR 17.961; *p* = 0.062) are associated with markedly increased odds of PIs, but none of these associations reach statistical significance.

## 4. Discussion

In our single-center, real-world cohort of patients with secondary AML treated with CPX-351, the incidence of PIs was 31.8%. This finding is notable and appears higher than the previously reported incidence in general acute leukemia populations, especially in AML [1]. However, the existing literature on this topic is dated, and research on infectious complications in AML has predominantly focused on other types of infections. We were surprised to find a high rate of PIs, given that CPX-351 is designed to reduce gastrointestinal side effects like mucositis, which are known to serve as an access point for pathogens and are considered a key risk factor for PIs [14]. A study from Renga et al. described how CPX-351 is able to provide protection for the gut from damage, inflammation, and dysbiosis by activating the aryl hydrocarbon receptor pathway. This activation represents a pivotal mechanism for preserving the integrity of the intestinal barrier and fostering a balanced gut ecosystem, as it promotes the production of protective cytokines like interleukin-22 and the anti-inflammatory cytokine interleukin-10 [15]. Even with the now extensive body of real-world literature on CPX-351, PIs are not specifically mentioned by previous studies [16]; instead, the focus remains on broader complications such as mucositis, febrile episodes, and septic shock. Our study, however, draws attention to the fact that while mucositis may be less frequent with CPX-351 (27% in our cohort versus 40% or more of standard chemotherapy in the literature [17]), attention must still be paid to infections of this type. In fact, although not statistically significant, there was a trend toward a higher incidence of PIs in patients with mucositis and a positive previous perianal swab, but the *p*-values were likely non-significant because of the limited number of events. In light of this, an optimal risk stratification strategy is warranted: this should include routine perianal inspection and symptom screening during the post-chemotherapy phase, especially in patients exhibiting mucositis, early imaging in cases of pain or swelling, and routine swab surveillance to guide the escalation to targeted therapies. In contrast to the general understanding that a longer duration of neutropenia leads to a higher incidence of infections, our analysis found no significant association between the duration of neutropenia and the onset of PIs, a finding that diverges from prior studies [2,18]. Despite both groups experiencing a long period of neutropenia, neither its duration nor the administration of G-CSFs was a significant factor in the development of PIs. Furthermore, while the restoration of the neutrophil count is typically associated with the improvement of the lesions [19], having severe neutropenia did not impede the tissue recovery, as the patient’s stable/improved/cured state at 30 days was independent of their neutropenia status. Effective management may have been determined by timely identification and early initiation of antibiotic treatment. The only factor that demonstrated a significant association with PIs in our cohort was the length of hospital stay, which was significantly longer in patients who developed a PI. As a matter of fact, this outcome is likely a consequence of the infection itself, requiring prolonged supportive care, antibiotic therapy, and in some cases, surgical intervention. Conversely, it is not possible to draw strong conclusions about the difference in terms of time to the next cycle, as other complications, such as coagulopathy and infections, in patients without a PIs may have similarly contributed to a prolonged inter-cycle interval. Crucially, the period between approximately 2020 and 2022 in Italy coincided with the major COVID-19 waves. This delay in hospital readmission could be primarily attributable to an increased risk of COVID-19 infections alongside a drastic reduction in hospital bed capacity caused by the redirection and saturation of hospital resources, leading to a severe shortage of availability. Moreover, the burden upon acute care and emergency departments was compounded by a substantial inflow of hematological patients presenting with sequelae related to post-COVID complications, thereby further constraining the already stressed infrastructure.

The limited and outdated literature on PIs in AML creates a significant knowledge gap, which is particularly relevant in the context of the escalating global challenge of antimicrobial multidrug resistance (MDR) [20]. In fact, while much of the existing research has focused on systemic infections like bacteremia and pneumonia, this approach may have overlooked a critical reservoir for these MDR pathogens: the patient’s own gut microbiome. The clinical utility of rectal swab surveillance is therefore crucial. Indeed, we observed a trend suggesting that a positive rectal swab prior to the onset of infection may be associated with the development of a PI. Notably, in the 2 cases with a positive pre-infection swab, the same bacterial species (Klebsiella pneumoniae) was later confirmed as the causative agent of the PI. While not statistically significant, likely owing to the small sample size, this observation highlights the value of routine surveillance and points to a potential link between the gut microbiome and subsequent PIs in this high-risk population. The presence of these specific pathogens in the rectal swabs suggests they may serve as a reservoir for subsequent infections [21]. Therefore, routine swab surveillance could be a valuable tool for identifying high-risk patients and potentially guiding preemptive or targeted prophylactic strategies.

With respect to the microbiological findings, Pseudomonas aeruginosa has been reported in older studies as the most frequently isolated bacterium from perianal lesions [1,22]. However, a more recent study has documented a variety of other prevalent species, including *Enterococcus* spp. and *Klebsiella* spp. [7], as was the case in our study.

Indeed, Enterobacteriaceae are the most frequently isolated germs in this context, confirming that the pathogens responsible for these infections originate from the patient’s own gut flora, rather than from the skin or soft tissue. Despite the high incidence of PIs, the outcomes in our cohort were encouraging. We observed a low rate of serious complications, with only 1 patient (14.2%) experiencing septic shock and no infection-related mortality or episodes of recurrence. Mortality rates for these infections range from 50% to 78% in the literature [23]. This wide range reflects differences in the groups of leukemia patients studied, including their other health conditions, chemotherapy plans, and the specific ways these infections were managed. Even if in our study none experienced a recurrence during subsequent chemotherapy cycles, based on existing research, the recurrence rate for PIs can be quite high, with some studies showing rates as high as 73% [24]. Therefore, preventative measures are crucial, and in light of this, preventing constipation is found to be very helpful [25] while the utility of procedures such as sitz baths or the topical application of antibiotics remains controversial [26]. Although antibiotic prophylaxis is often recommended for high-risk patients to prevent infections [27], our institutional policy is to avoid routine antibacterial prophylaxis specifically to mitigate the risk of developing MDR infections. Furthermore, its effectiveness in preventing recurrent anorectal infections has not been definitively established. Despite the lack of studies specifically designed to demonstrate the benefits of G-CSF as a prophylactic measure, evidence suggests that its administration is safe and does not negatively impact the quality of therapeutic response [6]. Indeed, our center’s policy has been to administer G-CSF to all patients as a prophylactic measure since the second wave of the Italian SARS-CoV-2 pandemic in October 2020. This approach is taken with the aim of reducing the duration of neutropenia and, consequently, mitigating the risk of infection.

Focusing on treatment, a conservative approach with antibiotic therapy was the primary strategy for all our patients. However, 3 patients (42.9%) underwent surgical drainage upon surgical consultation. A long-standing debate exists regarding the optimal management of PIs in neutropenic leukemia patients. Currently, there are no guidelines based on randomized studies that show whether surgical treatment is better than conservative treatment, or what the optimal timing for surgery might be. Switching from a broad-spectrum antibiotic to a targeted one as soon as the cultures and antibiogram results are available is a fundamental aspect of effective treatment. This practice is crucial because, while broad-spectrum antibiotics are necessary initially to cover all potential pathogens, their continued use could have significant negative consequences, such as altering the microbiome, leading to dysbiosis, and the selection of MDR species. Some studies advocate for a surgical approach [28,29], viewing it as the only truly definitive treatment for these conditions and arguing that it is equally effective in both immunocompromised and non-immunocompromised patients. In contrast, other research suggests that sole antibiotic therapy is a safer alternative. This conclusion is supported by the work of Carlson et al., who reported significantly higher mortality rates in the surgical group compared to those who underwent conservative therapy [30]. Still on the same topic, Grewal et al. conducted a retrospective study involving 81 patients to evaluate the outcomes of anorectal disease based on the type of treatment received. The findings indicated no statistically significant difference between the two treatment groups (operative versus non-operative) with regard to patient mortality and the rate of disease recurrence, suggesting that in this particular cohort, surgical intervention did not provide a measurable advantage over a non-operative approach for these crucial endpoints [31]. The existing body of literature on this topic is outdated. Consequently, a re-evaluation of the efficacy of antibiotic therapy compared to surgical intervention is needed, especially in light of the modern clinical challenge posed by multidrug-resistant organisms and the significant advancements in surgical techniques, which have made procedures safer and more effective. In our study, the surgical approach was well-tolerated, but it was associated with an increased mean length of hospital stay.

This extended hospitalization was an expected outcome, as it accounts for the longer recovery period and the additional care required following a surgical procedure. Our data therefore suggest that surgery can be a viable treatment option in a selected group of patients, but it comes with a trade-off in terms of a prolonged hospital stay. The decision to proceed with surgical intervention was guided by a combination of clinical symptoms and hematological criteria. Patient pain levels during hospitalization were quantitatively assessed using the NRS, a standard, direct measure of self-reported pain. A statistically significant difference was observed in mean NRS scores between the two patient groups (*p* = 0.001), with those undergoing surgery reporting a higher average pain score. This compelling clinical presentation, despite antibiotic therapy, appeared to be a primary factor in the decision to pursue a more aggressive approach. In addition to pain, the patients’ hematological profiles were critical to the decision-making process. To mitigate the risk of hemorrhage, a defining factor for surgical eligibility was the achievement of an adequate platelet count, specifically a threshold of greater than 20 × 10^9^/L. A statistically significant difference was found between the surgical and non-surgical groups regarding their ability to reach this threshold during hospitalization (*p* = 0.028). While the general trend for neutropenic patients was to await neutrophil count recovery before intervention, a specific exception was made. In one particularly complex case, a combination of intense pain, the size of the abscess, and a life-threatening clinical status compelled surgical drainage before the complete resolution of neutropenia. Ultimately, both the patients who underwent surgery and those who received a conservative approach demonstrated an acceptable short-term safety profile, with no 30-day mortality directly related to the infection or treatment procedures. These data provide evidence that both a non-invasive approach and a surgical intervention may be effective treatments, and the decision to intervene must be individualized according to the surgeon’s consultation and based on factors like general condition, thrombocytopenia, abscess extension, and neutrophil count.

A key strength of this study is its focus on a homogenous patient cohort, all treated with a standard induction regimen. This population also provides a unique opportunity to study outcomes in the absence of antibiotic prophylaxis, which can otherwise lead to bacterial resistance and the proliferation of pathogenic species. Nevertheless, the study has several limitations, including its single-center design, its retrospective nature, and the small sample size due to the rarity of secondary AML, which may have masked the effects of some variables, such as the duration of neutropenia. The current investigation provides essential descriptive data on the high incidence and characteristics of PIs in the CPX-351 cohort, but the lack of a contemporaneous comparator group remains a key limitation for inferring causality. To definitively determine whether the observed PI frequency is specifically related to the CPX-351 regimen or merely reflects a generalized risk inherent to AML patients, future research must prioritize comparative studies. Our critical next step will involve a comprehensive analysis comparing the incidence and severity of PIs in CPX-351-treated patients against those receiving conventional induction chemotherapies, which is necessary to identify risk factors leading to these infections, guide clinical prophylactic strategies, and find the optimal management strategy.

## 5. Conclusions

In conclusion, our study provides valuable, albeit limited, real-world data on the characteristics and outcomes of PIs in patients with secondary AML treated with CPX-351 in our center. The observed incidence rate is high and, if confirmed by larger-scale studies, would justify further investigation into the specific mechanisms and risk factors at play. Despite favorable clinical outcomes—with low rates of septic shock and no mortality—these infections are associated with considerable discomfort for the patient, along with the risk of extending the length of hospital stay to address these complications.

## Figures and Tables

**Table 1 cancers-18-00208-t001:** Patients’ characteristics.

	n = 22
Mean age, years ± SD (range)	58.8 ± 8.6 (43–72)
Sex (%)	
Female	8 (36)
Male	14 (64)
N. comorbidity (%)	
>4	2 (9)
4	2 (9)
3	5 (23)
2	9 (41)
1	2 (9)
0	2 (9)
Comorbidity Type (%)	
Cardiac disease	12 (21)
Renal disease	4 (7)
Respiratory disease	2 (3.5)
Hypercholesterolemia	4 (7)
Cancer	11 (19.5)
Others	24 (42)
Antifungal prophylaxis (%)	22.0 (100.0)
G-CSF post chemotherapy (%)	18.0 (81.8)
Antibacterial prophylaxis (%)	0 (0.0)
Parenteral nutrition (%)	11.0 (50.0)
Mucositis (%)	
Diarrhea	2.0 (9.1)
Oral mucositis	4.0 (18.2)
Positive rectal swab (%)	2.0 (9.1)
Total duration of neutropenia	
Mean, days ± SD (range)	35.5 ± 26.3 (0.0–100.0)
Length of hospital stay	
Mean, days ± SD (range)	41.5 ± 12.4 (21.0–75.0)
Days until start of next cycle	
Mean, days ± SD (range)	81.5 ± 21.5 (59.0–132.0)

N: number; SD: standard deviation; G-CSF: granulocyte colony-stimulating factor.

**Table 2 cancers-18-00208-t002:** Differences between surgery and non-surgical patients.

Surgery	Total (n = 7)	No (n = 4)	Yes (n = 3)	*p* Value
Mean age years ± SD (range)	57.0 ± 9.7 (46.0–72.0)	54.5 ± 12.0 (46.0–72.0)	60.3 ± 6.0 (54.0–61.0)	0.510 ^1^
N° neutrophils at onset of infection (×10^9^/L)				0.514 ^1^
Mean ± SD (range)	1.695 ± 3.814 (0–10.290)	2.612 ± 5.118 (10.0–10.290)	0.473 ± 0.662 (0–1.230)	
Total duration of neutropenia				0.539 ^1^
Mean, days ± SD (range)	30.2 ± 18.2 (0–50.0)	37.9 ± 29.6 (0–47.0)	30.3 ± 18.2 (18.0–50.0)	
Length of hospital stay				0.4 ^1^
Mean, days ± SD (range)	52.7 ± 15.6 (27.0–75.0)	47.5 ± 20.0 (42.0–75.0)	59.6 ± 0.58 (59.0–60.0)	
Mean NRS score ± SD (range)	6.29 ± 1.98 (4.0–9.0)	4.75 ± 0.5 (4.0–5.0)	8.33 ± 0.58 (8.0–9.0)	0.001 ^1^
Platelet count above 20 × 10^9^/L (%)	4.0 (57.2)	0	4 (100)	0.028 ^2^

N: number; SD: standard deviation; NRS: numeric rating scale. T-test ^1^, chi-squared test ^2^.

**Table 3 cancers-18-00208-t003:** Demographic and clinical characteristics of patients stratified by perianal infection.

	Total (n = 22)	No (n = 15)	Yes (n = 7)	*p* Value
Mean age years ± SD (range)	58.8 ± 8.6 (43–72)	59.7 ± 8.2 (43–72)	57.0 ± 9.7 (46–72)	0.510 ^1^
Gender				0.141 ^2^
Female	8.0 (36.4%)	7.0 (46.7%)	1.0 (14.3%)	
Male	14.0 (63.6%)	8.0 (53.3%)	6.0 (85.7%)	
Comorbidity	20.0 (90.9%)	14.0 (93.3%)	6.0 (85.7%)	0.563 ^2^
Total duration of neutropenia				0.539 ^1^
Mean, days ± SD (range)	35.5 ± 26.3 (0.0–100.0)	37.9 ± 29.6 (7.0–100.0)	30.3 ± 18.2 (0.0–50.0)	
G-CSF post chemotherapy	18.0 (81.8%)	13.0 (86.7%)	5.0 (71.4%)	0.388 ^2^
Length of hospital stay				0.034 ^1^
Mean, days ± SD (range)	41.5 ± 12.4 (21.0–75.0)	37.7 ± 8.3 (21.0–54.0)	49.6 ± 16.3 (21.0–75.0)	
Start next cycle				0.835 ^1^
Mean, days ± SD (range)	81.5 ± 21.5 (59.0–132.0)	80.7 ± 21.5 (59.0–132.0)	83.5 ± 24.6 (60.0–109.0)	
Parenteral nutrition	11.0 (50.0%)	6.0 (40.0%)	5.0 (71.4%)	0.361 ^2^
Mucositis	6.0 (27.3%)	2.0 (13.4%)	4 (57.2%)	0.053 ^2^
Positive rectal swab	2.0 (9.1%)	0	2.0 (28.6%)	0.090 ^2^

N: number; SD: standard deviation; G-CSF: granulocyte colony-stimulating factor. T-test ^1^, chi-squared test ^2^.

**Table 4 cancers-18-00208-t004:** Multivariate Firth logistic regression.

	Adjusted Odds Ratio (CI 95%, *p* Value)
Age (yrs)	1.036 (0.835–1.405, *p* = 0.709)
Gender (female/male)	6.339 (0.289–792.986, *p* = 0.214)
Comorbidity (no/yes)	0.209 (0.000–18.794, *p* = 0.494)
Total duration of neutropenia	0.990 (0.926–1.044, *p* = 0.653)
Parenteral nutrition (no/yes)	0.118 (0.001–2.098, *p* = 0.150)
Mucositis (no/yes)	17.961 (0.876–3472.276, *p* = 0.062)
Previous positive rectal swab (negative/positive)	5.554 (0.129–4861.502, *p* = 0.391)

CI: confidence interval.

## Data Availability

All relevant data are included in this article. For additional data inquiries, a request can be directed to the corresponding author.

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
