# Peer review of "Incidence and Characteristics of Perianal Infections in CPX-351-Treated AML Patients"

_cancers, 2026, doi:10.3390/cancers18020208_

Round 1
Reviewer 1 Report (Previous Reviewer 3)
Comments and Suggestions for Authors
In the current manuscript, the authors analyzed the incidence of perianal infections in patients with acute myeloid leukemia (AML) treated with CPX-351. The authors have provided several additional data and revised the manuscript in response to the reviewers’ previous comments. These changes have improved the clarity of the manuscript. However, there are still a few points that require further clarification.
-
In Table 1, please confirm whether the number “24” listed under “Others” in the comorbidity type category is correct.
-
Please add a more informative and appropriate title to Table 3 so that readers can more easily understand its content.
Author Response
Reviewer 1
In the current manuscript, the authors analyzed the incidence of perianal infections in patients with acute myeloid leukemia (AML) treated with CPX-351. The authors have provided several additional data and revised the manuscript in response to the reviewers’ previous comments. These changes have improved the clarity of the manuscript. However, there are still a few points that require further clarification.
- In Table 1, please confirm whether the number “24” listed under “Others” in the comorbidity type category is correct.
Yes, we confirm that the number is correct.
- Please add a more informative and appropriate title to Table 3 so that readers can more easily understand its content.
We are grateful for this suggestion. We modified accordingly and used the following title for Table 3: “Demographic and clinical characteristics of patients stratified by perianal infection.”

Reviewer 2 Report (Previous Reviewer 2)
Comments and Suggestions for Authors
Manuscript titled „Incidence and Characteristics of Perianal Infections in CPX-351-Treated AML Patients" written by Elisa Buzzatti et al. to evaluate the incidence and characteristics of PIs in a cohort of CPX-351-treated AML patients.
Although the topic is interesting, the manuscript still requires improvements.
Here are my major comments:
Please revise the abstract section in accordance with the changes indicated in the manuscript. The conclusions of the abstract should refer to the results of the present study rather than to comparisons with the existing literature. In addition, any results mentioned in the abstract should be supported by quantitative evidence (numerical data).
Author Response
Reviewer 2
Manuscript titled „Incidence and Characteristics of Perianal Infections in CPX-351-Treated AML Patients" written by Elisa Buzzatti et al. to evaluate the incidence and characteristics of PIs in a cohort of CPX-351-treated AML patients.
Although the topic is interesting, the manuscript still requires improvements.
Here are my major comments:
Please revise the abstract section in accordance with the changes indicated in the manuscript. The conclusions of the abstract should refer to the results of the present study rather than to comparisons with the existing literature. In addition, any results mentioned in the abstract should be supported by quantitative evidence (numerical data).
We thank the reviewer for the useful comment and we modified the abstract accordingly as follows:
“Background: Perianal infections (PIs) are a serious threat in patients with acute myeloid leukemia (AML). While CPX-351 is designed to reduce gastrointestinal toxicity, its impact on the incidence of PIs is unknown. This study aims to evaluate the incidence and characteristics of PIs in a cohort of CPX-351-treated AML patients.
Methods: We enrolled 22 adult patients diagnosed with secondary AML receiving CPX-351 between May 2020 and July 2025 at Policlinico Tor Vergata Hospital. Statistical analysis used descriptive statistics and multivariate analysis.
Results: The incidence of PIs in the cohort was 31.8%. Microbiological cultures from the lesions commonly yielded Klebsiella pneumoniae and Enterococcus species. The development of a PI was associated with a significantly longer hospital stay (mean, 49.6 vs. 37.7 days; p = 0.034). An increased odds ratio of having PIs was noted for mucositis and positive rectal swabs (17.961, p=0.062; 5.554, p=0.391 respectively), with two patients (28.5%) having a positive pre-infection swab for Klebsiella pneumoniae. Surgical intervention was guided by patient pain levels and hematological criteria. Surgical patients had significantly higher pain levels (p=0.001) and a platelet count greater than 20 x 10^9/L (p=0.028). All patients were alive at 30 days, with low rates of septic shock (14.2%, n=1) and no infection-related mortality or recurrence.
Conclusions: Despite CPX-351's known reduced gastrointestinal toxicity, our study showed a significantly higher incidence of PIs compared to literature data. While the outcomes were favorable, PIs led to prolonged hospitalization. Routine rectal swab surveillance could be a valuable tool for risk stratification and preemptive strategies.”

This manuscript is a resubmission of an earlier submission. The following is a list of the peer review reports and author responses from that submission.
Round 1
Reviewer 1 Report
Comments and Suggestions for Authors
E Buzatti & al submitted a manuscript about perianal infections in the context of CPX 351 treatment for patients diagnosed with secondary AML.
The use of CPX351 is generaly considered as more safe than a classical 3+7 for mucosal barriers. The question is not so clear for perianal infections. The question is of interst for the hematological community
Few comments:
1/ The use of surgery in this context is , in the majority of cases, to have an immediate effect on pain. Why in this study the surgery is delayed ? Neutrophils count and platelet count didn't delay this kind of surgery.
2/ The increase percentage of Perianal infections in this study for patients treated with CPX does not impact survival, and just delayed the discharge from hospital. But there is no difference in the time to next treatment. Why in this study the time to next treatment is more 80 days (More than two months ???)
Two minors comments: 1/ page 5 , line 1 and line 10, the sign +/- didn't appear in the text
Author Response
E Buzzatti & al submitted a manuscript about perianal infections in the context of CPX 351 treatment for patients diagnosed with secondary AML.
The use of CPX351 is generaly considered as more safe than a classical 3+7 for mucosal barriers. The question is not so clear for perianal infections. The question is of interst for the hematological community
Few comments:
1/ The use of surgery in this context is , in the majority of cases, to have an immediate effect on pain. Why in this study the surgery is delayed ? Neutrophils count and platelet count didn't delay this kind of surgery.
We thank the reviewer for raising this fundamental point. The optimal management strategy—antibiotic monotherapy versus combined surgical intervention—is a highly controversial and critical clinical challenge in this patient population. For hematological patients experiencing profound myelosuppression (evidenced by severely low platelet and neutrophil counts), surgical timing is perilous given the concurrent high risks of refractory sepsis and major hemorrhagic complications. In our study, surgical intervention was selectively performed only on patients who met acceptable pre-operative complete blood count parameters or those whose rapid clinical deterioration mandated immediate surgical management, despite the inherent risks. In the highly sensitive patient context of the hematological environment, surgeons are consistently apprehensive about performing operative procedures due to the significant risk of infectious complications and hemorrhagic events. This reluctance is particularly pronounced during the post-chemotherapy nadir, a period characterized by profound myelosuppression and compromised hemostatic capacity.
2/ The increase percentage of Perianal infections in this study for patients treated with CPX does not impact survival, and just delayed the discharge from hospital. But there is no difference in the time to next treatment. Why in this study the time to next treatment is more 80 days (More than two months ???)
We are grateful for giving us the opportunity to specify this point. After an induction chemotherapy, patients often experienced concurrent complications, such as coagulopathy and opportunistic infections, which ultimately prevented their re-admission to the hospital for the continuation of the therapeutic program. Crucially, the period between approximately 2020 and 2022 in Italy coincided with the major COVID-19 waves. This resulted in a significant challenge for hematological patients, many of whom experienced a prolongation of the inter-hospitalization interval. This delay was primarily attributable to an increased risk of COVID-19 infections alongside a drastic reduction in hospital bed capacity caused by the redirection and saturation of hospital resources, leading to a severe shortage of availability. Furthermore, the strain on emergency services was exacerbated by the influx of hematological patients presenting via the Emergency Department with post-COVID complications, which further intensified resource limitations. We have detailed this in the text.
Two minors comments: 1/ page 5 , line 1 and line 10, the sign +/- didn't appear in the text
We made the suggested corrections.
Reviewer 2 Report
Comments and Suggestions for Authors
Manuscript titled „Incidence and Characteristics of Perianal Infections in CPX-351-Treated AML Patients" written by Elisa Buzzatti et al. to evaluate the incidence and characteristics of PIs in a cohort of CPX-351-treated AML patients.
Although the topic is interesting, the manuscript requires improvements.
Here are my major comments:
Materials and Methods
- it would be helpful to include the ethics committee approval number in the Materials and Methods section.
- the term “incidence” may be misleading. It is not clear from the Materials and Methods section whether the study refers to new cases of PI. If this is not the case, the term ‘incidence’ is inappropriate and should be replaced with ‘frequency’ or ‘proportion
- Please specify which scale was used for the VAS. It does not seem to correspond to the one cited in the reference.
- Statistical analysis section: the authors declared that “Descriptive analyses were performed on the entire population and stratified by study group” but the study is not population-based — it relies on sampling. Therefore, the term ‘population’ is inappropriate. Please revise the wording accordingly.
- The Shapiro–Wilk test alone is not sufficient for assessing normality (please use QQ plots and other additional methods)
Results
- the Results section does not follow an appropriate scientific writing style and should be revised to ensure clarity, objectivity, and consistency.
- The authors state that “This study aims to evaluate the incidence and characteristics of PIs in a cohort of CPX-351-treated AML patients”. However, according to the Results section, only 22 secondary AML patients treated with CPX-351 (out of a total of 92 AML patients eligible for intensive chemotherapy during the study period) were included in the analysis. This represents a small, selected sample rather than a well-defined at-risk population. I would recommend clarifying that the study was based on CPX-351-treated secondary AML patients.
- In Table 1, dichotomous variables should be presented using only the category of interest (e.g., the ‘yes’ category), instead of listing both levels of the variable.
- Please reformulate the sentence: “Mean neutrophil count was 1.695 3.814 x 109/L (range 0- 10.290).” it is unclear and not correctly formatted.
- There are discrepancies between the results presented in the text and those reported in Table 2. (e.g. “Both groups experienced severe neutropenia as a consequence of chemotherapy, but the median duration of neutropenia did not significantly differ between groups (30.3 vs 37.9 days, p=0.54) but in Table 2, the authors reported the mean not median)
- Some paragraphs in the Results section include interpretative comments rather than data reporting (e.g. “The infection occurred during the neutropenic phase of chemotherapy and, due to severe pain and ineffective antibiotic therapy, surgical intervention was required. Drainage and seton placement were performed as soon as his neutrophil and platelet counts reached safe levels, leading to a complete recovery within 60 days”). These statements would be more suitably placed in the Discussion section.
- Table 3 does not represent a univariate analysis- please reformulate
Author Response
Manuscript titled „Incidence and Characteristics of Perianal Infections in CPX-351-Treated AML Patients" written by Elisa Buzzatti et al. to evaluate the incidence and characteristics of PIs in a cohort of CPX-351-treated AML patients.
Although the topic is interesting, the manuscript requires improvements.
Here are my major comments:
Materials and Methods
- it would be helpful to include the ethics committee approval number in the Materials and Methods section.
- We added the ethics committee approval number as suggested.
- the term “incidence” may be misleading. It is not clear from the Materials and Methods section whether the study refers to new cases of PI. If this is not the case, the term ‘incidence’ is inappropriate and should be replaced with ‘frequency’ or ‘proportion
- We thank the reviewer for the pertinent and accurate comment regarding our epidemiological terminology. We confirm that the use of the term “incidence” is appropriate for our study. We agree that clear definition is essential, and we apologize if the Materials and Methods section did not fully convey this point. To clarify: our analysis was specifically designed to measure the rate of newly developed cases of PI (Primary Infection) within our cohort over the defined observation period. Patients were confirmed to be free of PI at baseline (or at the initiation of the follow-up period) and were subsequently tracked to determine the number of individuals who developed the condition de novo. This methodology aligns precisely with the definition and calculation of cumulative incidence.
We will revise the Materials and Methods section to explicitly state that our calculation refers to the development of new PI cases during the follow-up time, thereby eliminating any potential ambiguity and justifying the retention of the term “incidence.
- Please specify which scale was used for the VAS. It does not seem to correspond to the one cited in the reference.
- Thank you for your valuable feedback regarding the pain assessment scale used in our study. Your comment is appropriate and highlights an important distinction in terminology.
We acknowledge that the classic Visual Analogue Scale (VAS) traditionally uses a continuous 100-millimeter (or 10-centimeter) line, where the score is measured in millimeters.
We apologize for the inaccuracy in our initial nomenclature. The scale we actually employed—where participants selected a whole number from zero (0) to ten (10) to indicate their pain intensity is correctly identified as the Numeric Rating Scale (NRS).
- Statistical analysis section: the authors declared that “Descriptive analyses were performed on the entire population and stratified by study group” but the study is not population-based — it relies on sampling. Therefore, the term ‘population’ is inappropriate. Please revise the wording accordingly.
- Thank you for your comment, we corrected accordingly.
- The Shapiro–Wilk test alone is not sufficient for assessing normality (please use QQ plots and other additional methods)
- We consulted with our statistician about the Shapiro-Wilk normality test, which was used because it is preferable when the sample size is less than 50. Reference: Mishra P, Pandey CM, Singh U, Gupta A, Sahu C, Keshri A. Descriptive statistics and normality tests for statistical data. Ann Card Anaesth 2019;22(1):67-72. Habibzadeh F. Data Distribution: Normal or Abnormal? J Korean Med Sci. 2024 Jan 22;39(3):e35.
Results
- the Results section does not follow an appropriate scientific writing style and should be revised to ensure clarity, objectivity, and consistency.
- We revised the writing style according to your suggestion.
- The authors state that “This study aims to evaluate the incidence and characteristics of PIs in a cohort of CPX-351-treated AML patients”. However, according to the Results section, only 22 secondary AML patients treated with CPX-351 (out of a total of 92 AML patients eligible for intensive chemotherapy during the study period) were included in the analysis. This represents a small, selected sample rather than a well-defined at-risk population. I would recommend clarifying that the study was based on CPX-351-treated secondary AML patients.
- We thank the reviewer for this valuable input. The specified information has been incorporated into the manuscript, and we have adjusted the relevant section accordingly to enhance clarity.
- In Table 1, dichotomous variables should be presented using only the category of interest (e.g., the ‘yes’ category), instead of listing both levels of the variable.
- Thank you for the comment, we modified the table accordingly.
- Please reformulate the sentence: “Mean neutrophil count was 1.695 3.814 x 109/L (range 0- 10.290).” it is unclear and not correctly formatted.
- Yes, thank you, the mistake was corrected in the updated version of the manuscript.
- There are discrepancies between the results presented in the text and those reported in Table 2. (e.g. “Both groups experienced severe neutropenia as a consequence of chemotherapy, but the median duration of neutropenia did not significantly differ between groups (30.3 vs 37.9 days, p=0.54) but in Table 2, the authors reported the mean not median)
- Yes, thank you, the mistake was corrected in the updated version of the manuscript.
- Some paragraphs in the Results section include interpretative comments rather than data reporting (e.g. “The infection occurred during the neutropenic phase of chemotherapy and, due to severe pain and ineffective antibiotic therapy, surgical intervention was required. Drainage and seton placement were performed as soon as his neutrophil and platelet counts reached safe levels, leading to a complete recovery within 60 days”). These statements would be more suitably placed in the Discussion section.
- We agree with the reviewer's suggestion. The descriptions of the three surgical cases have been rewritten to present only the objective clinical data, with all interpretive content relocated to the discussion section.
- Table 3 does not represent a univariate analysis- please reformulate
- Yes, thank you, the mistake was corrected in the updated version of the manuscript.
Reviewer 3 Report
Comments and Suggestions for Authors
In the current manuscript, the authors analyzed the incidence of perianal infections (PIs) in patients with acute myeloid leukemia (AML) who received CPX-351 during 2020–2025. The results are helpful for readers managing AML cases treated with CPX-351. However, several points require attention to further improve the manuscript.
- A comparison of the clinical characteristics between patients treated with CPX-351 and those treated with other induction regimens would provide deeper insight into whether CPX-351 is specifically associated with the development of PIs.
- While the authors compared baseline characteristics between patients with and without PIs, a formal risk analysis would be more informative for clinical practice. In particular, identifying predictors of PI development based on the available variables would enhance the clinical utility of the study. Consultation with a statistician may help strengthen this analysis.
- Several sentences appear to be redundant or repeated across the Introduction, Results, and Discussion sections. Reorganizing these sections to reduce overlap would improve readability and clarity for the audience.
Author Response
In the current manuscript, the authors analyzed the incidence of perianal infections (PIs) in patients with acute myeloid leukemia (AML) who received CPX-351 during 2020–2025. The results are helpful for readers managing AML cases treated with CPX-351. However, several points require attention to further improve the manuscript.
- A comparison of the clinical characteristics between patients treated with CPX-351 and those treated with other induction regimens would provide deeper insight into whether CPX-351 is specifically associated with the development of PIs.
Yes, certainly. We agree that a comparative analysis with patients treated with other induction regimens is critical for a deeper insight. However, we initially focused on meticulously describing the characteristics of these perianal infections (PIs) specifically within the CPX-351 cohort. This primary goal was motivated by the clinical paradox of observing such frequent gastrointestinal infections in these patients, who theoretically should have a lower incidence of such complications. Our future objective is indeed to conduct a follow-up study that will include patients treated with other chemotherapy regimens, allowing us to perform the necessary comparison and determine if the association between CPX-351 and PI development is statistically specific. We included in the text a paragraph describing the necessity for a comparisons with other chemotherapy regimens.
- While the authors compared baseline characteristics between patients with and without PIs, a formal risk analysis would be more informative for clinical practice. In particular, identifying predictors of PI development based on the available variables would enhance the clinical utility of the study. Consultation with a statistician may help strengthen this analysis.
We thank the reviewer for this essential comment. We fully agree that a formal risk analysis is necessary to determine the independent predictive value of the observed variables. We have since collaborated with our statistical expert to perform the requested analysis, which now allows us to more accurately identify patients at high risk. The revised manuscript reflects these statistical findings.
- Several sentences appear to be redundant or repeated across the Introduction, Results, and Discussion sections. Reorganizing these sections to reduce overlap would improve readability and clarity for the audience.
We thank the reviewer for their comment, and we have revised the writing accordingly.
Round 2
Reviewer 2 Report
Comments and Suggestions for Authors
From my point of view, the manuscript has not been sufficiently improved in accordance with the recommendations.
Reviewer 3 Report
Comments and Suggestions for Authors
The authors provided additional data and modified the manuscript based on the reviewers’ comments. These changes have certainly improved the clarity of the manuscript. However, several points still require attention to further strengthen the work.
-
Regarding the logistic regression analysis, the results are informative for helping readers understand the risk factors for PIs. However, the manuscript does not sufficiently explain how the authors selected variables for the model or how the logistic regression was actually performed. For example, the presence of mucositis was identified as a risk factor for PIs, but mucositis could be considered a subtype or component of PIs, which may confound the current risk analysis. In addition, it remains unclear how the authors incorporated the total duration of neutropenia into the model. Ideally, a multivariate analysis should be performed to more robustly evaluate the risk factors, although the number of PI cases is small. These methodological points should be discussed in greater detail.
-
In Table 1, please verify whether the number listed under “Comorbidity type – others” is correct. Additionally, does the current category of secondary AML include both therapy-related AML and AML transformed from MDS or MPN?